# What is the best strategy for iron deficiency anemia prevention and control in Iran? a policy analysis study protocol

Azadeh Dehghani[1,2], Maryam Rafraf[1]*, Fatemeh Mohammadi-Nasrabadi[3], Rahim Khodayari-zarnaq[4]

**1** Faculty of Nutrition and Food Science, Department of Community Nutrition, Nutrition Research Center, Tabriz University of Medical Sciences, Tabriz, Iran, **2** Student Research Committee, Tabriz University of Medical Sciences, Tabriz, Iran, **3** Faculty of Nutrition Sciences and Food Technology, Research Department of Food and Nutrition Policy and Planning, National Nutrition and Food Technology Research Institute, Shahid Beheshti University of Medical Sciences, Tehran, Iran, **4** Department of Health Policy and Management, School of Management and Medical Informatics, Tabriz University of Medical Sciences, Tabriz, Iran

* rafrafm@tbzmed.ac.ir, rafrafm2000@yahoo.com

**Data Availability Statement:** Deidentified research data will be made publicly available when the study is completed and published.

## Abstract

### Background

The present study is a type of exploratory qualitative studies and applied research. The approach of this study is a prospective policy analysis in which we will formulate appropriate policy options to prevent and control iron deficiency anemia in Iran.

### Methods and materials

Current study is a multi-method research with an analysis for policy approach containing three phases. First, through a literature review study, policies, programs and interventions of different countries to control and prevent anemia caused by iron deficiency will be identified. Then, in the qualitative phase of the study, the challenges, barriers, facilitators of the policies and programs implemented and ongoing in Iran will be examined. The content and policy-making process, as well as the context and role of stakeholders and actors will be analyzed using the framework of the policy triangle and analysis of the policy process using the Kingdon's multiple streams model. Then, the proposed initial policy options will be developed. In the next phase, an expert panel contain experts, authorities and policymakers will be formed and the proposed options will be reviewed and categorized. In order to prioritize policy options and evaluate their feasibility in Iran, the Delphi technique and the policy options analysis framework of the Centers for Disease Control and Prevention (CDC) will be used. At the end, policy options will be selected based on the highest score and will be presented as appropriate policy options.

### Conclusion

Prospective policy analysis allows the selection of potentially practical and effective policy options to control iron deficiency anemia. The findings of current study will be presented as reports and research articles for policy makers.

**Funding:** The Grant number 69140 was received by Maryam Rafraf from the Nutrition Research Center of Tabriz University of Medical Sciences, Tabriz, Iran. The funder only reviewed and approved the study proposal. The funders had no role in study design, data collection and analysis, decision to publish, or preparation of the manuscript.

**Competing interests:** The authors declare that there is no conflict of interests.

## Introduction

Iron deficiency anemia (IDA) is one of the most common micronutrient disorders worldwide, which affects the health, social, and economic well-being of millions of people in the world, including men, women and children [1,2]. Low bioavailability of dietary iron, decreased iron absorption and blood loss are possible causes of iron deficiency (ID) [3]. Anemia caused by chronic ID leads to cognitive and behavioral disorders in infants and children, fatigue and reduced work ability in older children and adults, prematurity and perinatal mortality in pregnant women [4,5]. Inadequate amounts of iron-rich foods, low content of iron in the diet, poor environmental sanitation, unsafe drinking water, iron loss due to parasite load (such as malaria or intestinal worms), repeated pregnancies in low-resource countries, and adolescent pregnancies are the main causes of a disproportionate increase in the prevalence of IDA [6,7]. Estimations in developing countries show that the economic consequences of IDA are very wide and according to the analysis of the World Health Organization (WHO) and the World Bank, it is the third main cause of lost life years due to illness, disability or death (Disability Adjusted Life Years (DALY)) in women of reproductive age [8,9].

Due to rapid socio-economic changes, lifestyle and food consumption, Iran is undergoing a nutritional transition. Changes in consumption and food pattern in recent decades have affected the status of micronutrients [10]. External and internal factors such as limited access to public health services, inadequate health knowledge and intolerance of distributed tablets have contributed to reducing the effectiveness of iron control programs [11–14]. Also, many of these interventions have been implemented using only semi-trained personnel to provide programs in different areas including schools [15].

In a meta-analysis study in 2017, the total prevalence of IDA and ID among the population aged 18 and less was 13.9% and 26.9% in Iran, respectively [16]. The Sayyari's survey on the prevalence of anemia was reported a 15% prevalence rate for anemia in children 2–12 years old in Iran [17]. Based on the WHO criteria (hemoglobin) Hb) level of $<12$ g/dl) a mild prevalence of anemia (7.9%) was existed in the northwest of Iran. A study revealed that the prevalence of IDA among girls and boys was 8.5% and 7.9%, respectively [18]. In a 2013 study in the rural areas of Tabas, central Iran, the prevalence of anemia among women of reproductive age was reported as 13.8% [19]. The prevalence of anemia was 19.0% in elderly people (aged 60 years and older) in Amirkola city in the north of Iran (20.3% in women and 17.9% in men) [20]. The highest prevalence was observed in some provinces of Iran like south of Khorasan, south of Kerman, Sistan and Baluchestan with average of 44% and the lowest prevalence were 9% in Kohgiloyeh, Boyer-Ahmad, Esfahan, and Yazd [21].

After holding a three-day meeting in 1995, which was held with the presence of representatives of WHO and UNICEF (United Nations International Children's Emergency Fund), several solutions were proposed to control and prevent anemia caused by ID. For the prevention and control of IDA, WHO announces to all countries a combination of four principal strategies, containing iron supplementation, proper nutrition education, food fortification with iron compounds, and the control of parasitic and infectious diseases. In the fortification strategy, the most common food carriers used in the world according to the studies are wheat, corn, salt, sugar and spices. In Iran, wheat flour was chosen because it is the dominant staple of the people [15,22–26]. The supplementation program was implemented as a pilot program in some provinces of Iran in 2001, and it was implemented as a national program throughout the country in 2005. Iron supplementation and education programs in the field of preventing and controlling ID and proper nutrition patterns are being implemented for different groups of society, including pregnant and lactating women, children under 2 years of age, and teenage girls of reproductive age [15,23,27–29]. Holding educational classes can lead to increase

awareness regarding intestinal parasitic diseases, improving mothers' nutritional behaviors, and increasing mother's awareness and attitude to control IDA [30–33].

Despite the availability of iron supplements and fortified foods in developed countries, their availability in developing countries is limited or often very expensive. Global, national and regional efforts to prevent and control ID and IDA have increased in recent years, but its prevalence does not seem to be reducing. Despite the efforts of the Ministry of Health and Medical Education (MOHME) of Iran in implementing the program of free iron supplements for pregnant and lactating women, children, and adolescent girls, more interventions are needed to augment the compliance and use of supplements. Food security and socioeconomic status are important factors, too [16,26].

Prospective policy analysis studies will authorize policymakers to analyze popular and common policies and suggest feasible and cost-benefit policy options to recuperate common conditions. Progressing study will be conducted to analyze and introduce policy options to prevent and control IDA in Iran. For current study, the following research questions have been specified:

1. What are the policies, strategies and programs of other countries to control and prevent anemia caused by ID and how effective are they?

2. What are the effective background factors and content of IDA prevention policies in Iran?

3. What is the process of IDA prevention policies in Iran?

4. What are the key stakeholders and their role in relation to IDA prevention policies in Iran?

5. What are the proposed policy options for preventing anemia caused by IDA in Iran?

6. What are the final options and their implementation capabilities?

## Methods and materials

Current study is a multi-method research with an analysis for policy approach containing three phases. The protocol of the current study was approved and registered by the Research Vice-Chancellor of Tabriz University of Medical Sciences, Tabriz, Iran (ethical code: IR. TBZMED.REC.1401.032; Grant number: 69140). The approach of this qualitative study is a case study; because by prospectively analyzing policies and formulating policy options to prevent IDA, we want to gain a deep understanding of these policies, programs, events, or related organizations. The current study will be carried out in three phases as follow (Fig 1).

**Phase 1**:

• **Reviewing international literatures and analyzing the policies, programs and strategies of other countries regarding the prevention and control of IDA:**

## Study design

In this specific purpose, we will review the policies, programs and strategies carried out and their impact in other countries for prevent anemia caused by ID. The content of IDA prevention policies in different countries of the world will be examined. The findings of this step will be useful in introducing policy options proper to the circumstances of the Iranian community.

## Sampling

Reducing the prevalence of IDA based on international standards will be considered as the main outcome of policy effectiveness. Considering that the general purpose of this research is

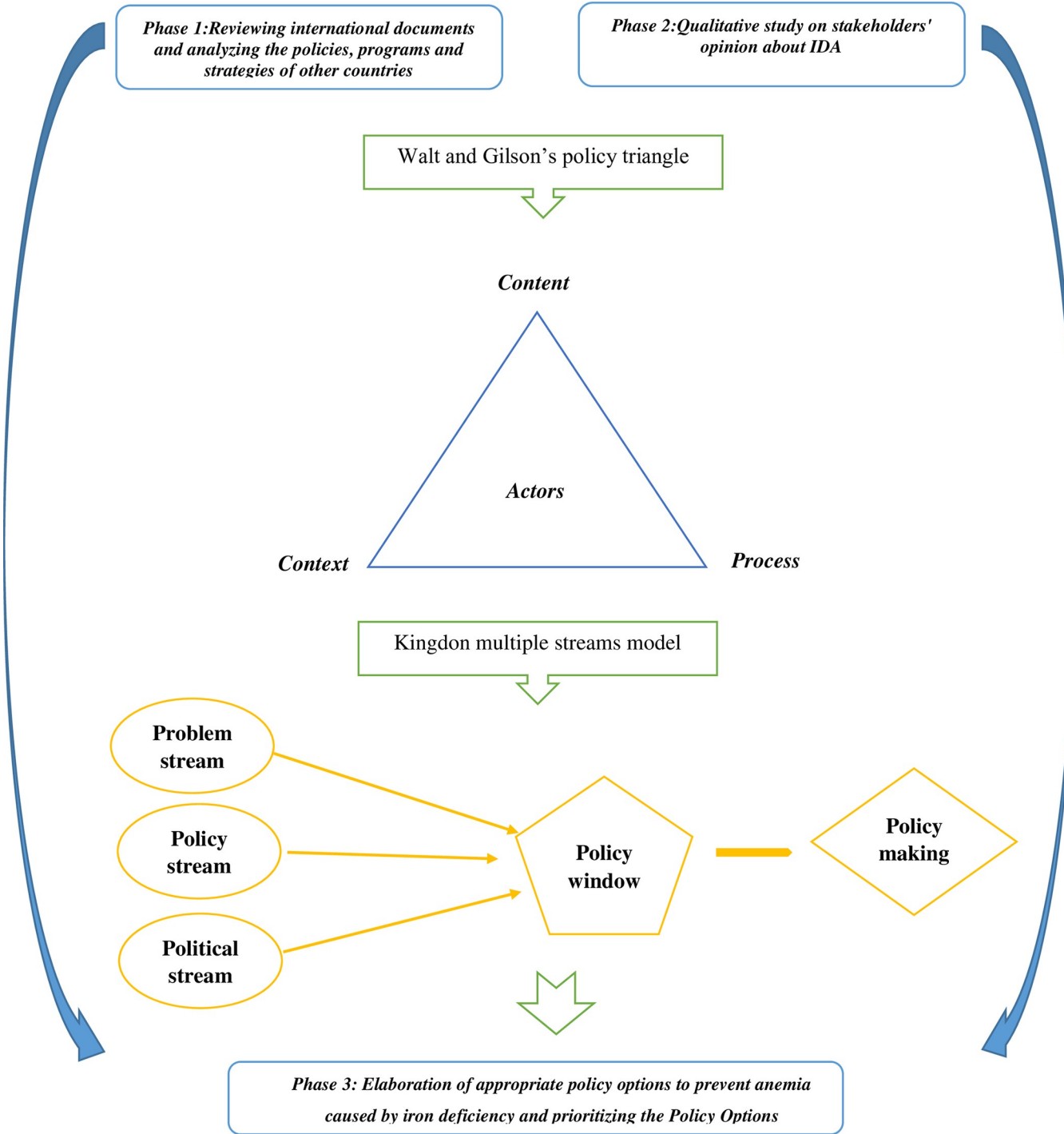

**Fig 1. Study flowchart on policy analysis and development upcoming policy options for the prevention and control of iron deficiency.**

to formulate policy options for preventing anemia caused by ID in Iran, it is possible to use the content of some international policies (according to their feasibility in Iran) as an input to formulate policies for Iran.

## Data collection and analysis

A data collection checklist based on critical infrastructure criteria will be developed to review successful programs. Then, the comprehensive systematic review of international documents will be conducted according to the Preferred Reporting Items for Systematic reviews and Meta-Analyses (PRISMA). Inclusion criteria include published studies on the prevention, screening, and control of IDA in the world, as well as studies that described at least one IDA prevention policy. Exclusion criteria also include clinical trials that have been conducted in patients with diseases affecting anemia such as kidney diseases. The MEDLINE, PubMed, ISI Web of Science, Scopus, WHO databases will be searched. A manual search of the references of the articles and reports entered will be also done. The following keywords will be used in the search:

(((('iron-deficiency anemia'[Title/Abstract] OR iron[Title/Abstract] OR anemia [Title/ Abstract] OR anaemia [Title/Abstract] OR ferritin [Title/Abstract] OR microcytic [Title/ Abstract]) AND ('nutrition'[Title/Abstract] OR 'nourish'[Title/Abstract] OR 'food'[Title/ Abstract] OR 'diet'[Title/Abstract] OR 'intake'[Title/Abstract] OR 'eat'[Title/Abstract] OR 'feed'[Title/Abstract] OR 'dish'[Title/Abstract] OR 'meal'[Title/Abstract] OR 'supplement'[Title/Abstract] OR 'fortification'[Title/Abstract] OR 'education' [Title/Abstract] OR 'Intestinal Parasite'[Title/Abstract])) AND ('policy'[Title/Abstract] OR 'program'[Title/Abstract] OR 'strategy'[Title/Abstract] OR 'intervention'[Title/Abstract] OR 'prevent'[Title/Abstract] OR 'screen'[Title/Abstract])) NOT ('randomized clinical trial'[Title/Abstract] OR 'randomized controlled trial'[Title/Abstract])

**Phase 2**:

- **Qualitative Study:**

## Study design

The second phase is a qualitative study with a conventional content analysis approach [34]. In this phase, three specific objectives will be assessed:

1. analyzing the context and content of IDA prevention policies in Iran

2. analyzing the process of IDA prevention policies in Iran

3. analysis of the stakeholders, their position and role and their connections with the prevention policies of anemia caused by ID in Iran.

## Sampling

The available policy documents will be searched through the website of the Ministry of Health and Medical Education, which can be easily obtained, and will be reviewed and analyzed for the content of IDA prevention policies in Iran. According to the review of documents and reports, key stakeholders, organizations, institutions, and key experts will be identified and interviewed. The policy documents and the key informants and stakeholders related to the prevention of IDA policy in Iran will be the research community at this stage. The document review form will be designed and used by the researcher, and then information such as title, document specifications/time of publication, place of publication, content of the document will be included in it. The snowball sampling method will be used and other stakeholders will be identified through informed people in the initial stage and will be added to the research samples [35]. Interviews will continue until data saturation, which means when the researcher

hears the similar sentence points iteration over and over without a new point or theme or notion emerging [36].

## Data collection and analysis

The interview will be designed by the researcher using the interview guide (Table 1). If possible, the interviews will be done face-to-face by the first author (AD) at a comfortable location agreed between the researcher and the participant. The method of data analysis in the current research will be the descriptive approach of conventional content analysis, so our assumption will descriptive saturation [37]. When no new descriptive codes, categories or themes emerged from the data analysis in the interviews, and based on our experiences [38], we will conclude that our descriptive data is reach saturation. Both of the richness and thickness of the data [39], will be emphasized in the present study, in addition to thick data, we try to have detailed, complex, multi-layered and delicate and high-quantity data by increasing the number of interviews so that we can interview all parts of the relevant systems involved. We also will conduct in-depth interviews with participants, thus ensuring the depth of the data [40]. In the initial stages, a few interviews will be conducted as a pilot and the necessary amendments will be applied in the interview guide based on the participants' comments for more clarity and precision.

The researcher will refer to the people of the relevant organizations and institutions by making an introduction letter and coordinating and setting a previous appointment. A written informed consent will be taken from each participant before interviewing. A tape recorder will be used to record the interviews and key points will be taken by the researcher. The duration of each interview will be 24 to 34 minutes and the interview will continue until data saturation. An audio recorder will be used to record all conversations during the interview. As a result, they will progressively focus on the proceeding data gathering, the expansion of the topics, and the process of developing of the developed themes [41]. To conduct qualitative content analysis for data analysis, all interviews will manually transcribe verbatim and read several times. The codes will inductively generate, and the extract codes will recognize as the categories based on the differences and similarities.

The framework of the policy triangle containing four components: context, content, process, and stakeholders, which is known as the Walt and Gilson triangle, will be used for the

**Table 1. The principal interview guide regarding IDA in Iran.**

| Questions |
|---|
| 1. Give your opinion from various cultural, social, and health dimensions regarding the prevention policies of IDA in Iranian society? |
| 2. When and why did IDA prevention become one of the health policy priorities in the country? What are the factors influencing the agenda of IDA prevention and control? |
| 3. What are the upcoming challenges and opportunities and the weaknesses and strengths of Iran's health system in the management and prevention of IDA? |
| 4. What are the challenges and problems that exist in the phase of developing policies related to IDA? |
| 5. What are the challenges and problems that exist in the implementation phase of policies related to IDA? |
| 6. What are the challenges and problems that exist in the consequence and evaluation phase of policies related to IDA? |
| 7. What is the success rate or fulfillment of IDA prevention goals? What are the obstacles, threats, opportunities, challenges and positive and negative points of each policy? |
| 8. What organizations and institutions are the main stakeholders and actors in the field of IDA? Which ones are more influential? |
| 9. What is the role of scientific evidence and documentation regarding IDA prevention policies in the country? |
| 10. What are your suggestions and implementation strategies for the success of the program? |

analysis of health sector policies [42]. This framework can be done by creating analysis diagrams based on the theme or based on the case or a combination of both. Framework analysis is used in three phases: description, analysis phase, and interpretation phase. The description stage includes applying the interviews and recalling essential points during the interview, the analysis will include the classification of the text of the interviews and documents. The interpretation stage will include the process of coding and theming.

Analysis of the policy process in the ordering stage will be done using the Kingdon's multiple streams model and in the stage of formulation, implementation and evaluation of IDA prevention policies, the policy cycle model will be used [43]. MAXQDA version 2020 software is used for coding and analyzing qualitative data. We will use Policy maker and MAXQDA software to perform stakeholder analysis, too. At this stage, in order to obtain information about the interests, positions and abilities of the stakeholders, a checklist will be prepared including the role of each stakeholder in each activity and the sources of power of the stakeholder in the two aspects of the policy maker and the policy implementer. The power and position of the actors will be obtained from the Policy maker software. Immediately after the interview, the stakeholders' responses to each question will be entered into the stakeholder profile table in the software.

**Phase 3**:

- **Elaboration of appropriate policy options to prevent anemia caused by ID and prioritizing the policy options**

## Study design

In the last phase we will assess two specific objectives:

1. Formulating suitable policy options to prevent anemia caused by ID for Iran

2. Prioritizing and checking the feasibility of proposed policy options and preparing a policy brief to prevent anemia caused by ID in Iran.

## Sampling

At this phase, by summarizing the results obtained from the review study, the policy analysis, the semi-structured interview of the actors, and the analysis of the stakeholders, the barriers and facilitators of the policies for the prevention and control of anemia caused by ID are identified and the policy options extracted by the researchers will be categorized and organized and then the initial draft of policy options will be compiled. The level of support, the approach, perspective and position of the stakeholders and the use of their power and facilities will be discussed, and the level of participation of an organization and stakeholder (high, medium, low and no participation) will also be examined. In the next step, the situational, structural, cultural and environmental factors that can have a direct and indirect effect on the IDA in the society will be included and a SWOT analysis will be done to identify the strengths, weaknesses, opportunities and threats.

## Data collection and analysis

The extracted options will be examined in the expert panel consisting of relevant experts and policy makers. The suggested opinions of the panel members will be noted and collected. The main experts and policymakers who must be present in the meetings will discuss about the feasibility, acceptability and political support of the options. Based on the results of the study, the

proposed options will be reviewed and priority solutions will be selected [44]. A checklist will be compiled to check the feasibility and acceptability and will be provided to the panel members. Panel meetings will continue until saturation is reached. After the last meeting of the panel, all opinions and suggestions regarding the options will be summarized and the options will be prepared. In order to prioritize policy options and assess their feasibility in Iran, the Delphi technique [45,46] and the policy options analysis framework of the Centers for Disease Control and Prevention (CDC) will be used [47]. The CDC framework has three main criteria: 1- degree of impact on public health 2- feasibility and 3- economic and budgetary impact. The final version of the formulated policy option will be sent to each member present in the panel meetings and after prioritization, the final options will be presented based on priority. Eventually, selected options will be ranked applying to prepare the upmost suitable policy options in the current situation for Iran.

## Study status

The study protocol was approved by the medical ethics committee of the Tabriz University of Medical Sciences. The first phases of the study are in progress. So far, an umbrella systematic review and meta-analysis related to the prevalence of IDA has been carried out and published using the Preferred Reporting Items for Systematic reviews and Meta-Analyses (PRISMA) checklist in order to determine the current IDA situation in our country [48]. Moreover, a review of international and national documents is underway. The interviews are currently being performed, too.

## Discussion

The protocol will provide a comprehensive and precise description of the strategies used to evaluate the implementation of IDA control intervention policy. The study will provide rigorous and essential evidence on which governments, policy makers and other organizations can develop strategies to combat and prevent IDA.

By using policy analysis, it is ensured that a systematic process is carried out to opt the ideal option for the going situation in the studied setting. The data from all stages of current study will be triangulated, which will provide us with a useful and practical set of data to create a comprehensive perspective. The views of key stakeholders, senior managers, policy makers, knowledgeable actors and staff, expert service recipients as well as studying successful implemented programs in other communities will be used to investigate community-based self-governing policies and obtain the most suitable policy options derived from the context of the community and will present them to the country's policy makers to promote evidence-informed policy making.

Some studies such as Aghapour et al.'s study in the field of Vitamin D deficiency prevention [49], Rajaeieh et al. in the prevention of vitamin A deficiency [50], Fathi et al., regarding policies to promote physical activity [51], Taghizadeh et al., regarding the prevention of childhood obesity [52,53], and Kabiri et al., regarding the prevention of gastrointestinal cancer [54], use the same approach. Also, in some review articles, the use of this approach has been considered in studies [55]. They have been able to provide good recommendations to Iranian politicians. It is expected that this study can provide useful and effective recommendations in the field of iron deficiency anemia.

Another type of qualitative studies whose results can be used by experts, decision makers, managers and other stakeholders to focus on key issues and identify areas of interest is SWOT analysis. A SWOT analysis evaluated the burden of chronic diseases across Europe and assess what policies and programs to address diabetes? This qualitative study has been carried out

using a SWOT analysis on policy/programme addressing NCDs applicable at the national or subnational level. The SWOT analyses produce a general picture of the complexity of designing and implementing effective programmes and policies that meet to local needs [56]. A qualitative research study employed a grounded theory approach that was conducted in a rural area of Indonesia with the aim of investigating the key aspects of an integrated health care system model to prevent IDA in adolescent girls. The results showed that the model of the integrated health care system includes several essential points, these points include the awareness of policymakers and the efforts of teenage girls, and the support of parents, teachers and society [57].

A mixed-methods cross-sectional study was conducted to examine the knowledge, attitudes, and understanding of mothers and health care workers in Lebanon, as well as to examine the main determinants of anemia among Syrian refugee children living in multiple crisis situations. The results showed that in order to reduce the high burden of anemia among children, multi-part interventions of medical and financial support in combination with nutrition counseling for mothers are needed [58].

Jafari et al to investigate the barriers and facilitators of iron supplementation program among adolescent females, conducted a cross-sectional study by random cluster sampling method on 399 high school girls from North, South, East, West and Central regions of Iran using a researcher-made questionnaire. The results of their research showed attention to different aspects of the program, including providing better quality iron supplements, designing more attractive educational programs, providing more suitable environmental conditions, employing more experienced executive staff and strengthening public notification is necessary to achieve the goals of the program [59].

Prieto-Patron et al.'s study investigated reducing the burden of IDA in Ivory Coast through Wheat flour and condiment fortification using a comparative risk assessment model. The results of the study showed that fortification helped reduce the IDA burden by approximately 5%. Since the Ivory Coast is one of the places with high prevalence of malaria and other infectious diseases, food fortification can be an effective nutritional intervention to prevent and control infectious diseases. Therefore, the findings of this study provide additional input to policy makers in the IDA area regarding the extent of impact and thus can support the concept of future fortification strategies [60]. Zakariah-Akoto's et al study examined experiences and perceptions of anaemia prevention, and iron-folic acid (IFA) supplementation usage among pregnant women and antenatal for boosting the supplementation programme by utilizing a qualitative approach (6 focus group and 10 in-depth interviews with care providers at two hospitals in Accra, Ghana). The results of their study showed that food insecurity and misconceptions about supplementation should be addressed as part of attempts to address low adherence to supplementation and high rates of maternal and child anemia in Ghana [61].

In addition, focusing on the prerequisites and barriers to the prevention and control of IDA among people at each stage of the program implementation process and changing the practice will aid us to create more suitable and compatible policies for the society. The results of this study will provide policy options to address the challenges of IDA prevention and control in the country's current conditions at both the local and national levels. Moreover, it can be served as a basis for creating wide national and international discourses to make more practical strategies to prevent IDA among all age groups of the society.

## Conclusion

Current study will assign scientific evidence and documents of the barriers to the implementation of IDA controlling policies in Iran by analyzing policies in the elements of content,

context, stakeholders and process by applying the Policy Triangle model. It is hoped the Iranian health policymakers consider the policy options whenever IDA controlling programs among Iranian community. Moreover, the results of this study will display the challenges of the IDA policymaking cycle, and the obstacles to the implementation of existing policies in this field. Our study will represent the literature with utmost proper policy options based on the criteria of acceptability, feasibility and effectiveness for controlling IDA at the local and national levels. Findings will also create a foundation for broader international discourse to expand strategies compliant with national and local structure and function and their challenges for controlling IDA.

## Acknowledgments

We gratefully acknowledge the financial support of the Nutrition Research Center of Tabriz University of Medical Sciences, Tabriz, Iran (Grant number: 69140).

## Author Contributions

**Conceptualization:** Azadeh Dehghani, Maryam Rafraf, Fatemeh Mohammadi-Nasrabadi, Rahim Khodayari-zarnaq.

**Data curation:** Azadeh Dehghani.

**Methodology:** Azadeh Dehghani, Rahim Khodayari-zarnaq.

**Project administration:** Azadeh Dehghani, Fatemeh Mohammadi-Nasrabadi.

**Supervision:** Maryam Rafraf.

**Validation:** Azadeh Dehghani.

**Writing – original draft:** Azadeh Dehghani.

**Writing – review & editing:** Maryam Rafraf, Fatemeh Mohammadi-Nasrabadi, Rahim Khodayari-zarnaq.

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
