## [Decision Letter · Decision Letter 0]

19 Jun 2024

PONE-D-23-37892What is the best strategy for iron deficiency anemia prevention and control in Iran? A policy analysis study protocolPLOS ONE

Dear Dr. Rafraf,

Thank you for submitting your manuscript to PLOS ONE. After careful consideration, we feel that it has merit but does not fully meet PLOS ONE’s publication criteria as it currently stands. Therefore, we invite you to submit a revised version of the manuscript that addresses the points raised during the review process.

We look forward to receiving your revised manuscript.

Kind regards,

Abdu Oumer

Academic Editor

PLOS ONE

 [The Grant number 69140 was received by MR from the Nutrition Research Center of Tabriz University of Medical Sciences, Tabriz, Iran. The funder only reviewed and approved the study proposal. 

https://pazhoohan.tbzmed.ac.ir/].  

4. PLOS requires an ORCID iD for the corresponding author in Editorial Manager on papers submitted after December 6th, 2016. Please ensure that you have an ORCID iD and that it is validated in Editorial Manager. To do this, go to ‘Update my Information’ (in the upper left-hand corner of the main menu), and click on the Fetch/Validate link next to the ORCID field. This will take you to the ORCID site and allow you to create a new iD or authenticate a pre-existing iD in Editorial Manager. Please see the following video for instructions on linking an ORCID iD to your Editorial Manager account: https://www.youtube.com/watch?v=_xcclfuvtxQ.

Additional Editor Comments (if provided):

Reviewers' comments:

Reviewer's Responses to Questions

**Comments to the Author**

1. Does the manuscript provide a valid rationale for the proposed study, with clearly identified and justified research questions?

Reviewer #1: Yes

Reviewer #2: Yes

2. Is the protocol technically sound and planned in a manner that will lead to a meaningful outcome and allow testing the stated hypotheses?

Reviewer #1: Yes

Reviewer #2: Yes

3. Is the methodology feasible and described in sufficient detail to allow the work to be replicable?

Reviewer #1: No

Reviewer #2: Yes

4. Have the authors described where all data underlying the findings will be made available when the study is complete?

Reviewer #1: No

Reviewer #2: Yes

5. Is the manuscript presented in an intelligible fashion and written in standard English?

Reviewer #1: No

Reviewer #2: Yes

6. Review Comments to the Author

You may also provide optional suggestions and comments to authors that they might find helpful in planning their study.

Reviewer #1: PONE-D-23-37892

What is the best strategy for iron deficiency anemia prevention and control in Iran? A policy analysis study protocol

This manuscript presents an interesting research protocol to formulate policy options to prevent and control iron deficiency anemia (IDA) in Iran. To do so, the authors planned a three-phase method. The first phase comprises a literature review on IDA policies in other countries, the second phase consists of a review of existing IDA policy documents in Iran and interviews with stakeholders, organizations, and institutions to identify challenges, barriers, and facilitators of ongoing policies and programs implemented in Iran. The third and last phase consists of the development of policy options for Iran. Then these options will be presented to an expert panel composed of relevant policymakers that will, after analysis, provide a priority list among the options, with a discussion about its feasibility, acceptability, and political support. It is expected that the expert panel find agreement on the most suitable policy for the current situation of IDA in Iran.

Major issues

Language editing is recommended.

1. On page 6, lines 148-166, the authors could clarify: What is the type of review conducted is it systematic, scope, narrative, or other? Was a review protocol registered in any platform, such as PROSPERO, for example? Was any guideline used to help plan the review (e.g. PRISMA)?

2. In the ‘Sampling’ section of page 7, lines 177/178 show a phrase that states that ‘[…] all the key informants and stakeholders related to the prevention of IDA policy in Iran will be the research community’. The authors could clarify how they will be able to identify ‘all’ key informants and stakeholders, will there be a method to select and identify who is key and who is not? Besides, the phrasing is rather confusing. In line 178, the authors write that ‘all available policy documents will be reviewed and analyzed […]’. How will the authors be able to select ‘all’ available policy documents in Iran? Is there a government database containing these documents, will the authors run another literature review?

3. On page 8, lines 194/195, the authors could provide more information on how they will determine they had enough interviews for the pilot study before performing any necessary amendments or modifications.

4. On page 8, line 198, the author refers to the use of a tape recorder. Clarification would be beneficial on how the recordings will be used, if there will be a transcript, and if so, will the transcription be made manually or if a software will be used.

Minor issues

1. Lines 97/98 show a phrase that says ‘The supplementation program in Iran was implemented as a pilot program in some provinces of Iran in 2001, and it will be implemented as a national program throughout the country in 2005.’ Timing seems mistaken.

2. Lines 155/156 describe the method in a future tense; however, line 257 says this phase of the study is in ultimate stage.

3. Line 193: readers would benefit if reference to tables and figures numbers were embedded in the text.

4. Line 241: what do the authors mean when saying, ‘after determining the priority options, they will be finalized’. What does finalized mean in this context? Does it mean that these priority options will be selected as the result of the study? The reader would benefit from clarification.

Reviewer #2: Dear author,

Thank you for the nice work and I do have two comments. The result and discussion is very shollow and please discuss in detail by comparing similar study published else were. Update the old references.

7. PLOS authors have the option to publish the peer review history of their article (what does this mean?). If published, this will include your full peer review and any attached files.

Reviewer #1: No

Reviewer #2: **Yes: **Habtamu Fekadu Gemede

---

## [Author Response · Author response to Decision Letter 0]

30 Jul 2024

Reviewer #1: PONE-D-23-37892

What is the best strategy for iron deficiency anemia prevention and control in Iran? A policy analysis study protocol

This manuscript presents an interesting research protocol to formulate policy options to prevent and control iron deficiency anemia (IDA) in Iran. To do so, the authors planned a three-phase method. The first phase comprises a literature review on IDA policies in other countries, the second phase consists of a review of existing IDA policy documents in Iran and interviews with stakeholders, organizations, and institutions to identify challenges, barriers, and facilitators of ongoing policies and programs implemented in Iran. The third and last phase consists of the development of policy options for Iran. Then these options will be presented to an expert panel composed of relevant policymakers that will, after analysis, provide a priority list among the options, with a discussion about its feasibility, acceptability, and political support. It is expected that the expert panel find agreement on the most suitable policy for the current situation of IDA in Iran.

Major issues

Language editing is recommended.

1. On page 6, lines 148-166, the authors could clarify: What is the type of review conducted is it systematic, scope, narrative, or other? Was a review protocol registered in any platform, such as PROSPERO, for example? Was any guideline used to help plan the review (e.g. PRISMA)?

Thank you for your heedful comment. Language editing was done and following descriptions were added to the relevant section:

A data collection checklist based on critical infrastructure criteria will be developed to review successful programs. Then, the comprehensive systematic review of international documents will be conducted according to the Preferred Reporting Items for Systematic reviews and Meta-Analyses (PRISMA).

2. In the ‘Sampling’ section of page 7, lines 177/178 show a phrase that states that ‘[…] all the key informants and stakeholders related to the prevention of IDA policy in Iran will be the research community’. The authors could clarify how they will be able to identify ‘all’ key informants and stakeholders, will there be a method to select and identify who is key and who is not? Besides, the phrasing is rather confusing. In line 178, the authors write that ‘all available policy documents will be reviewed and analyzed […]’. How will the authors be able to select ‘all’ available policy documents in Iran? Is there a government database containing these documents, will the authors run another literature review?

Thank you for your vigilant comment. Following descriptions were added to the relevant section:

The available policy documents will be searched through the website of the Ministry of Health and Medical Education, which can be easily obtained, and will be reviewed and analyzed for the content of IDA prevention policies in Iran. According to the review of documents and reports, key stakeholders, organizations, institutions, and key experts will be identified and interviewed. The policy documents and the key informants and stakeholders related to the prevention of IDA policy in Iran will be the research community at this stage.

3. On page 8, lines 194/195, the authors could provide more information on how they will determine they had enough interviews for the pilot study before performing any necessary amendments or modifications.

Thank you for your careful comment. Following descriptions were added.

The interview will be designed by the researcher using the interview guide (Table 1). If possible, the interviews will be done face-to-face by the first author (AD) at a comfortable location agreed between the researcher and the participant. The method of data analysis in the current research will be the descriptive approach of conventional content analysis, so our assumption will descriptive saturation (37). When no new descriptive codes, categories or themes emerged from the data analysis in the interviews, and based on our experiences(38), we will conclude that our descriptive data is reach saturation. Both of the richness and thickness of the data(39), will be emphasized in the present study, in addition to thick data, we try to have detailed, complex, multi-layered and delicate and high-quantity data by increasing the number of interviews so that we can interview all parts of the relevant systems involved. We also will conduct in-depth interviews with participants, thus ensuring the depth of the data(40).

4. On page 8, line 198, the author refers to the use of a tape recorder. Clarification would be beneficial on how the recordings will be used, if there will be a transcript, and if so, will the transcription be made manually or if a software will be used.

The following clarification was added based on the reviewer’s comment:

A tape recorder will be used to record the interviews and key points will be taken by the researcher. The duration of each interview will be 24 to 34 minutes and the interview will continue until data saturation. An audio recorder will be used to record all conversations during the interview. As a result, they will progressively focus on the proceeding data gathering, the expansion of the topics, and the process of developing of the developed themes (41). To conduct qualitative content analysis for data analysis, all interviews will manually transcribe verbatim and read several times. The codes will inductively generate, and the extract codes will recognize as the categories based on the differences and similarities.

Minor issues

1. Lines 97/98 show a phrase that says ‘The supplementation program in Iran was implemented as a pilot program in some provinces of Iran in 2001, and it will be implemented as a national program throughout the country in 2005.’ Timing seems mistaken.

Thank you for your careful comment, the sentence has been corrected.

The supplementation program was implemented as a pilot program in some provinces of Iran in 2001, and it was implemented as a national program throughout the country in 2005.

2. Lines 155/156 describe the method in a future tense; however, line 257 says this phase of the study is in ultimate stage.

Tense of verbs were modified based on the reviewer’s comments.

3. Line 193: readers would benefit if reference to tables and figures numbers were embedded in the text.

Thank you for your thoughtfulness. It was done.

4. Line 241: what do the authors mean when saying, ‘after determining the priority options, they will be finalized’. What does finalized mean in this context? Does it mean that these priority options will be selected as the result of the study? The reader would benefit from clarification.

The following explanation was added to clarify for the readers based on the reviewer’s comment:

The extracted options will be examined in the expert panel consisting of relevant experts, policy makers. The suggested opinions of the panel members will be noted and collected. The main experts and policymakers who must be present in the meetings will discuss about the feasibility, acceptability and political support of the options. Based on the results of the study, the proposed options will be reviewed and priority solutions will be selected(44).

Reviewer #2: Dear author,

Thank you for the nice work and I do have two comments. The result and discussion is very shollow and please discuss in detail by comparing similar study published else were. Update the old references.

Thank you for encouraging comment. Some explanations were added to Study status and comparing with similar studies was done in Discussion. Some old references were updated; however, others are related to the historical perspectives (meetings and guidelines) of iron deficiency anemia prevention and control and could not be updated.

Another type of qualitative studies whose results can be used by experts, decision makers, managers and other stakeholders to focus on key issues and identify areas of interest is SWOT analysis. A SWOT analysis evaluated the burden of chronic diseases across Europe and assess what policies and programs to address diabetes? This qualitative study has been carried out using a SWOT analysis on policy/programme addressing NCDs applicable at the national or subnational level. The SWOT analyses produce a general picture of the complexity of designing and implementing effective programmes and policies that meet to local needs (56). A qualitative research study employed a grounded theory approach that was conducted in a rural area of Indonesia with the aim of investigating the key aspects of an integrated health care system model to prevent IDA in adolescent girls. The results showed that the model of the integrated health care system includes several essential points, these points include the awareness of policymakers and the efforts of teenage girls, and the support of parents, teachers and society(57). 

A mixed-methods cross-sectional study was conducted to examine the knowledge, attitudes, and understanding of mothers and health care workers in Lebanon, as well as to examine the main determinants of anemia among Syrian refugee children living in multiple crisis situations. The results showed that in order to reduce the high burden of anemia among children, multi-part interventions of medical and financial support in combination with nutrition counseling for mothers are needed(58).

Jafari et al to investigate the barriers and facilitators of iron supplementation program among adolescent females, conducted a cross-sectional study by random cluster sampling method on 399 high school girls from North, South, East, West and Central regions of Iran using a researcher-made questionnaire. The results of their research showed attention to different aspects of the program, including providing better quality iron supplements, designing more attractive educational programs, providing more suitable environmental conditions, employing more experienced executive staff and strengthening public notification is necessary to achieve the goals of the program (59).

Prieto-Patron et al.'s study investigated reducing the burden of IDA in Ivory Coast through Wheat flour and condiment fortification using a comparative risk assessment model. The results of the study showed that fortification helped reduce the IDA burden by approximately 5%. Since the Ivory Coast is one of the places with high prevalence of malaria and other infectious diseases, food fortification can be an effective nutritional intervention to prevent and control infectious diseases. Therefore, the findings of this study provide additional input to policy makers in the IDA area regarding the extent of impact and thus can support the concept of future fortification strategies (60). Zakariah-Akoto's et al study examined experiences and perceptions of anaemia prevention, and iron-folic acid (IFA) supplementation usage among pregnant women and antenatal for boosting the supplementation programme by utilizing a qualitative approach (6 focus group and 10 in-depth interviews with care providers at two hospitals in Accra, Ghana). The results of their study showed that food insecurity and misconceptions about supplementation should be addressed as part of attempts to address low adherence to supplementation and high rates of maternal and child anemia in Ghana (61).

---

## [Editor Report · Decision Letter 1]

17 Sep 2024

What is the best strategy for iron deficiency anemia prevention and control in Iran? A policy analysis study protocol

PONE-D-23-37892R1

Dear,

We’re pleased to inform you that your manuscript has been judged scientifically suitable for publication and will be formally accepted for publication once it meets all outstanding technical requirements.

Kind regards,

Abdu Oumer

Academic Editor

PLOS ONE
---

## [Editor Report · Acceptance letter]

24 Sep 2024

PONE-D-23-37892R1 

PLOS ONE

Dear Dr. Rafraf, 

I'm pleased to inform you that your manuscript has been deemed suitable for publication in PLOS ONE. Congratulations! Your manuscript is now being handed over to our production team.

Kind regards, 

on behalf of

Dr. Abdu Oumer 

Academic Editor

PLOS ONE